# Epigenetic Evolution of ACE2 and IL-6 Genes: Non-Canonical Interferon-Stimulated Genes Correlate to COVID-19 Susceptibility in Vertebrates

**DOI:** 10.3390/genes12020154

**Published:** 2021-01-25

**Authors:** Eric R. Sang, Yun Tian, Laura C. Miller, Yongming Sang

**Affiliations:** 1Department of Agricultural and Environmental Sciences, College of Agriculture, Tennessee State University, 3500 John A. Merritt Boulevard, Nashville, TN 37209, USA; ericr.sang@gmail.com (E.R.S.); ytian@tnstate.edu (Y.T.); 2Virus and Prion Diseases Research Unit, National Animal Disease Center, USDA-ARS, Ames, IA 50010, USA; laura.miller@usda.gov

**Keywords:** COVID-19, angiotensin converting enzyme 2, interferons, IL-6, epigenetic regulation

## Abstract

The current novel coronavirus disease (COVID-19) has spread globally within a matter of months. The virus establishes a success in balancing its deadliness and contagiousness, and causes substantial differences in susceptibility and disease progression in people of different ages, genders and pre-existing comorbidities. These host factors are subjected to epigenetic regulation; therefore, relevant analyses on some key genes underlying COVID-19 pathogenesis were performed to longitudinally decipher their epigenetic correlation to COVID-19 susceptibility. The genes of host angiotensin-converting enzyme 2 (ACE2, as the major virus receptor) and interleukin (IL)-6 (a key immuno-pathological factor triggering cytokine storm) were shown to evince active epigenetic evolution via histone modification and *cis*/*trans*-factors interaction across different vertebrate species. Extensive analyses revealed that ACE2 ad IL-6 genes are among a subset of non-canonical interferon-stimulated genes (non-ISGs), which have been designated for their unconventional responses to interferons (IFNs) and inflammatory stimuli through an epigenetic cascade. Furthermore, significantly higher positive histone modification markers and position weight matrix (PWM) scores of key *cis*-elements corresponding to inflammatory and IFN signaling, were discovered in both ACE2 and IL6 gene promoters across representative COVID-19-susceptible species compared to unsusceptible ones. The findings characterize ACE2 and IL-6 genes as non-ISGs that respond differently to inflammatory and IFN signaling from the canonical ISGs. The epigenetic properties ACE2 and IL-6 genes may serve as biomarkers to longitudinally predict COVID-19 susceptibility in vertebrates and partially explain COVID-19 inequality in people of different subgroups.

## 1. Introduction

First identified in Wuhan, China, December 2019, the novel coronavirus disease 2019 (COVID-19) spread worldwide and caused over 0.68 million confirmed deaths and 17 million infected cases across 200 countries by the end of July 2020 [1,2]. COVID-19 stands out as a new zoonotic disease caused by Severe Acute Respiratory Syndrome coronavirus 2 (SARS-CoV-2) [3], which, in the view of a virus, obtains an effective balance between its deadliness and contagiousness in humans [4,5]. In line with that, patients with ages over 45, especially 75 years old, had a worse prognosis and 5–10-fold higher mortality rate than younger ones at 0–17 years old, who mostly showed a mild disease or even asymptomatic [6,7,8,9,10,11,12,13,14,15]. Similarly, higher mortality rates were observed in males than females, and particularly in the patients who have pre-existing medical conditions (comorbidities) regardless of gender or age [6,7,8,9,10,11,12,13,14,15]. These underlying comorbidities include diabetes, cancer, immunodeficiency, hypertension and cardiovascular disease, asthma and lung disease, kidney disease, as well as chronic gastrointestinal (GI)/liver disorders. In addition to predictable symptoms of cough, fever and headache from the lung infection, the virus can spread to almost every organ including the brain, heart, gut, kidneys, and skin to cause organ-specific problems [6,7,8,9,10,11,12,13,14,15]. Therefore, on the host side, SARS-CoV-2 susceptibility and disease progression of COVID-19 is a phenomenon of epigenetic regulation, which underlies the diversity of the disease progression throughout the body system and across different patients that share a near-identical genetic background, except those which have inborn genetic mutations [16,17,18,19].

Zoonosis and reverse zoonosis infer a dynamic exchange of pathogens between humans and animals, particularly domestic and wild vertebrates. This constitutes a major challenge for both public health and animal health, and unites them into one ecological health. The potential infection of SARS-CoV-2 in both wild and domestic animals is a public health concern after the COVID-19 prevalence in human society [20,21]. This concern emphasizes: (1) the identification of reservoir animal species that originally passed SARS-CoV-2 to humans; and (2) potential risks of infected people passing the virus to animals, particularly domestic species, to form an amplifying zoonotic cycle and exacerbate SARS-CoV-2 evolution and cross-species transmission [20,21]. Studies have provided evidence that domestic minks, cats, and dogs could be virally or serologically positive for SARS-CoV-2 [20,21,22,23,24,25,26,27,28], as were several Bronx Zoo tigers [29]. Experimental animal inoculations with human SARS-CoV-2 isolates demonstrated that ferrets, hamsters, domestic cats, and some non-human primate species were susceptible to human SARS-CoV-2 strains; however, pigs, alpacas, and (putatively) cattle are not [20,21,22,23,24,25,26,27,28,29]. Previously, we and several others have proposed structural simulation models of ACE2 and the viral S-Receptor binding domain (S-RBD) to predict SARS-CoV-2 susceptibility across representative vertebrates, especially major domestic and wild mammalian species [30,31,32,33]. The structural affinity between ACE2 and S-RBD plays a primary role in the viral attachment and accessibility in cells, and the specific early cellular responses that regulate ACE2 expression and signal early immune responses determine the host susceptibility to the virus [34,35,36,37,38,39,40]. We propose an integrative model, which incorporates both ACE2-RBD structural affinity (primarily determined by cross-species genetic difference) and epigenetic regulation of key genes during the early phase of the virus-host interaction, to predict host COVID-19 susceptibility and disease progression [30,31,32,33].

Among the core host factors that determine COVID-19 susceptibility and early disease progression, angiotensin-converting enzyme 2 (ACE2) and interleukin (IL)-6 were focused upon because of their critical roles directly involved in viral infection and host immunopathies [41,42,43,44,45]. In SARS-CoV-2 pathogenesis, ACE2 serves as the primary receptors for cell attachment and entry [42,43]. Several groups have reported that SARS-CoV-2 exerts higher receptor affinity to human ACE2 than other coronaviruses, which may contribute to the high-contagiousness and rapid spread of SARS-CoV-2 in humans [42,43]. Being a key enzyme in the body’s renin–angiotensin–aldosterone system (RAAS), ACE2 catalyzes angiotensinogen (AGT) to produce the active forms of hormonal angiotensin (Ang) 1–9, which directly regulate the blood volume/pressure, body fluid balance, sodium and water retention, as well as co-opt multiple effects on inflammation, apoptosis, and generation of reactive oxygen species (ROS) [43,44,45]. In this regard, not only do the virus direct binding and functional impairment of ACE2 enzymatic function, but also epigenetic regulation of ACE2 expression in various tissues/conditions, serve as a physio-pathological mechanism underlying the COVID-19 disease complex, and further relate to blood clotting, aneurisms and chilblains in infant patients [43,44,45,46].

SARS-CoV-2 seizes ACE2 for cell entry, which can be followed by a cytokine-related syndrome, namely acute respiratory distress syndrome (ARDS). Plausibly, the occupancy of the ACE2 catalytic domain by the viral spike protein (S) blocks AGT activation into Ang1–9 and leads to the accumulation of Ang2 in the serum [43,44,45,46]. Circulatory increase in Ang2 induces inflammatory cytokines, including TNF-α, IL-6, and soluble IL-6 receptor α (sIL-6Rα) in pneumocytes and macrophages, through binding Ang1-receptor (AT1R) and activating disintegrin- and metalloprotease 17 (ADAM17)-mediated cascade [41,42,43,44,45,46]. This process is followed by activation of the IL-6 amplifier (IL-6-AMP), which co-activates NF-κB and transcription factor STAT3 to enhance inflammatory response and leads to ARDS underlying COVID-19. Ang2-AT1R activation also induces pyroptosis, a highly inflammatory form of programmed cell death accompanying cytotoxicity caused by viral infections [41,45,46]. Aggregately, SARS-CoV-2 itself also activates NF-κB via various pattern recognition receptors (PPRs) [33,34,35,36,37,38,39,40]. Therefore, IL-6 and IL-6 AMP are biomarkers of hyperactivation of inflammatory machinery exacerbated by ACE2 blocking and viral infection, which represent key cytokines in deciphering cytokine-related syndrome and disease progression of COVID-19 [41,45,46].

The expression of ACE2 is inter-regulated by multiple physio-pathological factors, including intracellular pathogenic infection, pre-existing inflammatory condition from comorbidities, and inflammatory cytokines including TNF and IFNs [41,42,43,44,45,46]. Several studies have demonstrated that human *ACE2* gene behaved like an interferon-stimulated gene (ISG) and was stimulated by viral infection and IFN treatment; however, mouse *Ace2* gene was not [47,48,49]. Canonical ISGs describe over a thousand cellular genes that are induced by IFN simulation via the IFN-JAK-STAT signaling axis [50]. These canonical ISGs are mainly induced by type I and type III IFNs but overlap with those upregulated by type II IFN (i.e., IFN-γ) [47,48,49,50]. These ISGs comprise a front line of antiviral immunity to restrict virus spreading from the initial infection sites [50]. However, based on gene evolution and epigenetic analyses, ACE2 may not be a member of these classical antiviral ISGs, and more likely belongs to the non-canonical ISGs (non-ISGs) such as IL-6 (a.k.a. IFN-β2 in humans) [47,48,49,50,51]. These non-ISGs are primed under a pre-inflammatory condition and stimulated by IFN or IFN plus TNF through an epigenetic cascade involving positive histone modification (mainly H3K4me3 and H3K27ac) to increase chromatin accessibility for binding by transcription factors (including PU.1, IRFs, and NF-κB) and culminating in non-ISGs expression (Figure 1) [51,52,53,54]. To confirm that, we conducted cross-species comparative analysis between IL-6 and ACE2 genes. First, annotation of ENCODE epigenetic datasets discovered similarity of H3K4me3 and H3K27ac markers between IL-6 and ACE2 gene promoters in both humans and mice; however, significantly higher Z-scores and enrichment of H3K4me3 and H3K27ac in human IL-6 and ACE2 genes were detected than in their mouse orthologs, respectively [55]. Secondly, detection of *cis*-regulatory elements (CREs) that bind core transcription factors of non-ISGs, including PU.1, IRFs, and NF-κB, in ACE2 and IL-6 gene proximal promoter regions across 25 representative animal species [30,56]. Thirdly, we found that the evolutionary increase in ACE2, and especially the IL-6 gene response to inflammatory and IFN signaling may serve as an epigenetic marker for COVID-19 susceptibility in some animal species, including humans. Finally, using our non-biased RNA-Seq data, we further categorized more non-ISGs that resemble the expression pattern of either IL-6 or ACE2 [57]. Notably, we detected two ACE2 isoforms, which differ in both proximal promoters and coding regions, in some livestock species including pigs, dogs, and cattle [30]. In pigs, the ACE2 short isoform (ACE2S) has an expression pattern more similar to IL-6 than the long isoform (ACE2L). Collectively, our findings characterize ACE2 and IL-6 genes as non-ISGs responding differently to inflammatory and IFN signaling, and their epigenetic properties may serve as biomarkers to predict COVID-19 susceptibility in vertebrates longitudinally and partially explain COVID-19 inequality in people of different subgroups [20,30,31,32,33].

## 2. Materials and Methods

### 2.1. Annotation of ENCODE Epigenetic Datasets

The profile of epigenetic markers relevant to histone positive modification, mainly H3K4me3 and H3K27ac, were searched using the gene symbols through the ENCODE public domain at https://www.encodeproject.org/ under the default condition [55]. The ENCODE datasets for generating the epigenetic results include those mainly based on Chip-Seq and ATAC-Seq from 839 and 157 cell/tissue types of humans and mice, respectively. The Max Z-Scores and locations of the histone markers on the gene promoter regions were then curated under a permission for academic users, and manually diagramed.

### 2.2. Promoter Sequence Extraction and Alignment

The DNA sequences of the proximal promoters of analyzed genes were extracted from NCBI Gene and relevant databases (https://www.ncbi.nlm.nih.gov/gene). Both IL-6 and *ACE2* genes and corresponding transcripts have been well annotated in most representative vertebrate species. In most cases, the annotations were double verified through the same Gene entries at Ensembl (https://www.ensembl.org). The protein and DNA sequences were collected from all non-redundant transcript variants and further verified for expression using relevant RNA-Seq data (NCBI GEO profiles) (Appendix A). The proximal promoter region spanned ~2.5 kb before the predicted transcription (or translation) start site (TSS). The protein and DNA sequences were aligned using the multiple sequence alignment tools of ClustalW or Muscle through an EMBL-EBI port (https://www.ebi.ac.uk/). Other sequence management was conducted using programs at the Sequence Manipulation Suite (http://www.bioinformatics.org). Sequence alignments were visualized using Jalview (http://www.jalview.org) and MEGAx (https://www.megasoftware.net). Sequence similarity calculations and plotting were conducted using SDT1.2 (http://web.cbio.uct.ac.za/~brejnev). Other than those indicated, all programs were run with default parameters [30].

### 2.3. Examining Transcription Factor Binding Sites in the Gene Promoters and PWM Scoring

We used two programs/databases to confirm each other for the major CRE predictions. The regulatory elements (and corresponding binding factors) in the ~2.5 kb proximal promoter regions were examined against both the human/animal TFD Database using a program Nsite (Version 5.2013, at http://www.softberry.com). The mean position weight matrix (PWM) of key *cis*-elements in the proximal promoters were calculated using PWM tools through https://ccg.epfl.ch/cgi-bin/pwmtools, and the binding motif matrices of examined TFs were extracted from MEME-derived HOCOMOCOv11 TF collection affiliated with the PWM tools [56]. The species-specific CRE sequences were then extracted from each promoter sequence for alignments, as shown in Figure 3.

### 2.4. Phylogenic Analysis and Topological Comparison

Evolutionary analyses were conducted in MEGA X as described [30]. The evolutionary history was inferred by using the maximum likelihood method and Tamura–Nei model. Initial tree(s) for the heuristic search were obtained automatically by applying Neighbor-Join and BioNJ algorithms to a matrix of pairwise distances estimated using the Tamura–Nei model, and then selecting the topology with superior log likelihood value. For topological comparison between phylogenic trees generated using IL-6 and ACE2 gene proximal promoters, the phylogenies of Newick strings were generated using the MEGA program, and topological comparison between the Newick trees was performed with Compare2Trees at (http://www.mas.ncl.ac.uk/~ntmwn/compare2trees) to obtain the overall topological scores. Other than those indicated, all programs were run with default parameters as the programs suggested.

### 2.5. RNA-Seq and Data Analysis

During cross-species annotation of ACE2 and IL-6 genes, RNA-Seq datasets that are affiliated to NCBI gene entries (such as BioProjects PRJEB4337 and PRJNA66167 for humans and mice genes) were used to verify the gene expression per RNA-Seq exon/intron coverage analyses. The detailed records for NCBI RNA-Seq data analyses are provided as in the Appendix A. For expression confirmation, several sets of RNA-Seq data from NCBI Gene databases, and one of ours generated from porcine alveolar macrophages (BioProject with an accession number of SRP033717), were analyzed for categorizing ISGs and non-ISGs accordingly to the expression patterns of IL-6 and ACE2 genes. Significantly and differentially expressed genes (DEGs) between two treatments were described using an edgeR package and visualized using bar charts (RPKM) or heatmaps (Log_2_ fold ratio) as previously described [57].

## 3. Results and Discussion

### 3.1. Epigenetic Processes in Induction of Non-Canonical IFN-Stimulated Genes (Non-ISGs)

Studied mostly in humans and mice, the hundreds of classical ISGs, such as ISG15 and IRF1, contain the main IFN-responsive CREs, including IFN-stimulated regulatory element (ISRE) and γ-activated sequence (GAS), in their promoter regions [47,50]. The tripartite IFN-stimulated gene factor 3 (ISGF3), which is composed of three transcription factors including STAT1, STAT2 and IRF9, is activated downstream of the IFN-JAK-STAT signaling axis to bind ISREs and stimulate canonical ISG expression [47,50]. In addition to this classical axis to induce ISGs, IFNs also co-opt multiple non-canonical signaling pathways to activate these ISGs or other corresponding genes together through various alterative mechanisms [51,52,53,54]. These non-canonical IFN signaling pathways involve extensive crosstalk between the signaling pathways mediated by various cellular pathogen pattern-recognition receptors (PRRs) and inflammatory cytokines, notably IL-1, IL-6, and TNF [51,52,53,54]. The non-canonical signaling pathways not only diversify mechanisms for inducing ISGs, but also extend the spectrum of IFN-responsive genes, indicating a multifunctional property of IFNs in antiviral and immuno-physiological regulation [50,51,52,53,54]. Studies have shown that human IL-6 and ACE2 are two candidates for these non-ISGs [47,48,49,50,51]. Figure 1 shows the current understanding of the gene activation cascade of human IL-6 (and plausibly ACE2) genes as an example of non-ISGs, whose IFN-inductive property and systemic role have been recognized as underlying multiple inflammatory comorbidities [51,54]. In brief, stimulation of epithelial cells and tissue macrophages by early pro-inflammatory signaling of TNF induces transient expression of TNF-target genes encoding inflammatory mediators, such as IL6 and TNF. This is followed by a transient state that is insensitive to further inflammatory signaling from TLR activation, and thus relevant chromatin-containing non-ISGs are not activated (depicted by a grey shade in Figure 1). This transient suppression state, however, can be activated by a co-stimulation with TNF plus IFN-α, resulting in an increase in positive histone marks (H3K4me3 and H3K27ac) and chromatin accessibility of the gene promoter regions, which sequentially recruit the binding of corresponding transcription factors including IRFs and NF-κB to activate non-ISG expression [51,54]. Besides IL-6, many tunable ISGs, including human ACE2 as has been demonstrated, show sustainable responses to IFN and pathogenic inflammatory signaling, and share expression patterns involving epigenetic sensation and synergistic IFN-induction as depicted for non-ISGs (Figure 1) [47,48,49,50,51,52,53,54]. However, the cross-species evolutionary characterization of non-ISGs has not been studied. Using IL-6 and ACE2 as examples, extensive epigenetic and expression analyses were performed in this study to determine their epigenetic evolution and potential role as biomarkers to predict the susceptibility and disease progression of COVID-19.

### 3.2. Determine Species-Specific Positive Histone Marks in Human and Mouse IL-6 and ACE2 Gene Promoters

Epigenetic positive histone modification in a certain chromatin region, mainly including histone H3 with tri-methylation at the 4-lysine residue (H3K4me3) or with the acetylation at the 27-lysine residue (H3K27ac) here, is associated with a higher activation status of adjacent gene transcription, thus defined as positive epigenetic marks enhancing relevant gene expression. The enrichment of H3K4me3 and H3K27ac defines one epigenetic feature of non-ISGs post activation [51,52,53,54,55]. Through annotation of Chip-Seq and ATAC-Seq datasets from 839 and 157 cell/tissue types of humans and mice through ENCODE (https://www.encodeproject.org/) [55], we detected significant and comparative existence of H3K4me3 and H3K27ac markers between IL-6 and ACE2 gene promoters in various humans and mouse samples (Figure 2). However, higher Z-scores and enrichment of H3K4me3 and H3K27ac were found in human IL-6 and ACE2 genes (Figure 2A,B) than their mouse orthologs (Figure 2C,D). In both distal and proximal regions of the ACE2 gene promoters, the human gene (Figure 2A) was marked by 2–3-fold more positive histone modifications than the mouse ortholog, indicating higher activation and transcription activity of human IL-6 and ACE2 genes under similar conditions. Notably, human IL-6 is a short gene located distantly from other coding genes and might be correlated to higher histone marks (esp. H3K27Ac) relevant to distal super enhancers. By contrast, human ACE2 gene is a relatively long gene surrounded by other genes, and its major promoter is compacted in a more proximal region and has a limited H3K27Ac marks spanning the 2000–5000 bp distal region. Because these findings are extracted from the extensive datasets representing systemic sample types, it is convincing that typical epigenetic positive histone modifications, H3K4me3 and H3K27ac, are significantly associated with the promoter regions of ACE2 as with IL-6 genes. Specifically, IL-6 genes were shown to have more histone modifications around their proximal promoter regions than ACE2 genes, which had more in a very distal region (>20 kb). There were higher Z-scores and enrichment of these positive histone markers in the human genes than their mouse orthologs, indicating evolutionary and probably species-specific manners of epigenetic regulation of these non-ISGs [51,54]. This epigenetic difference of key non-ISGs might contribute to disease susceptibility and progression when animals of different species are exposed to same pathogenic pressure.

### 3.3. Cross-Species Comparison of Key Cis-Regulatory Elements (CREs) That Mark Non-ISG Regulation in IL-6 and ACE2 Genes

After determination of positive histone markers along the IL-6 and ACE2 gene bodies, we examined the existence of *cis*-regulatory elements (CREs) that interact with typical non-ISGs transcription factors, including PU.1 (a.k.a. SPI1), IRFs, and NF-κB1/2 in the promoter regions of IL-6 and ACE2 gene orthologs [50,51,52,53,54,55,56]. We extracted the primary promoter sequences from IL-6 and ACE2 genes from 25 representative vertebrate species, which contained ten previously validated SARS-CoV-2-susceptible species and other naturally unsusceptible species based on collected evidence [26,27,28,29,30]. As shown in Figure 3, all three types of CREs (i.e., PU.1, IRFs, and NF-κB) that mark non-ISG expression were mapped for cross-species existence in the promoter regions of both IL-6 and ACE2 genes. Significant PWM scores (*p* < 0.0001) were determined for their CREs when each was compared with the corresponding human CRE matrix (Figure 3A–C) [56]. ACE2 genes had a generally lower PWM score for these CREs than those for IL-6 genes, in particular the PWM scores for NF-κB2 CRE in ACE2 genes were at 2–8 Log_2_ units lower (Figure 3D). This indicates that ACE2 genes were less responsive to non-canonical NF-κB signaling mediated by NF-κB2 [58,59]. Because dysregulation of non-canonical NF-κB signaling contributes to various autoimmune and inflammatory diseases, the differential role of ACE2 and IL-6 in inflammatory immunopathies are worth further investigation [58,59]. Notably, only CRE matrices to IRF1 are shown in Figure 3C; both ACE2 and IL-6 gene promoters actually contain CREs binding IRF2-8 with high PWM scores, except for CREs interacting with IRF5 and IRF9 which had low PWM scores in most tested species (Figure 4 and Figure 5). Because IRF9 is a key component of ISGF3 and binding to ISREs to activate canonical ISG expression, this discovery evidently distinguishes ACE2 and IL-6 genes from the classical ISGs such as ISG15 and IRF1 (Figure 4) [50,51]. However, IL-6 genes of eight species maintain their IRF9 binding CREs; for example, in Zebrafish and frogs, only rat ACE2 gene showed a high PWM score for containing an IRF9 binding CRE (Figure 4). This further postulates a species-dependent trend of non-ISG evolution, and warrants further investigation in contributing to host–pathogen interactions.

Figure 5 gathers cross-species analyses of mean PWM scores of the CREs, which bind STAT1/2, PU.1 (a.k.a. SPI1), NF-κB1, NF-κB2, and multiple IRFs (including, IRF1-4, IRF7, and IRF8 that show significant PWM scores with *p* < 0.0001 under the algorithm’s default) in the proximal promoter regions of IL-6 and ACE2 gene orthologs from the 25 representative vertebrate species. As shown, these bookmarking CREs for non-ISGs had comparable Log_2_(mPWM) scores between ACE2 and IL-6 genes across different species, and also showed species-specific variation to some extent. IL-6 genes generally had higher mPWM scores for more of the tested animal species with CREs that bind STAT1/2 and IRFs downstream of IFN signaling (Figure 5A,E) [50,52]. Of significant difference between ACE2 and IL-6 genes was their CREs’ PWM scores pertinent to NF-κB1 and NF-κB2 (Figure 5C,D). Whereas ACE2 genes evolved to be slightly more responsive to the canonical NF-κB1 signaling in most mammalian species (Figure 5C), IL-6 genes obtained much higher responsiveness to non-canonical NF-κB2 signaling (Figure 5D). Studies have shown that defects in non-canonical NF-κB2 signaling are associated with severe immune deficiencies, and dysregulation of this pathway contributes to the pathogenesis of various autoimmune and inflammatory diseases [58,59]. The epigenetic difference of IL-6 and ACE2 genes downstream of canonical NF-κB1 and non-canonical NF-κB2 signaling thus may serve as differential gene markers for inflammatory-related syndromes [58,59].

### 3.4. Epigenetic Evolution of Higher PWM Scores of Non-ISG’s Core CREs in ACE2 and Especially IL-6 Gene Promoters in COVID-19-Susceptible Species

As previously described, in addition to its core role in the physiological regulation of blood volume/pressure and body fluid balance, the RAAS also critically affects inflammation, apoptosis, and other immune reactions. For instance, suppression of ACE2 increases Ang2 production to signal pro-inflammatory and apoptotic responses in affected tissues [44,45,46]. When exacerbated by infection of an intracellular pathogen, such as SARS-CoV-2 in COVID-19 cases, a high inflammatory form of programed cell death, known as pyroptosis, is induced, accompanying massive production of pro-inflammatory cytokines including IL-1, IL-6, TNF and CXCL10 [41,45,46]. Due to the potential clinical relevance to these CREs in COVID-19, we performed a comparative study to determine if the COVID-19 susceptible animal species obtained some epigenetic features in these core CREs in regulation of IL-6 and ACE2 expression. Figure 6 compares the mPWM scores of these core non-ISG CREs between two groups: known SARS-CoV-2-susceptible species [CoV2(+)] and unsusceptible species [CoV2(−)]. Figure 6 shows that ACE2 and IL-6 genes from CoV2(+) species contain CREs that have significantly higher mPWM scores. This indicates that, in some vertebrate species, non-ISGs such as ACE2 and especially IL-6 genes evolve to obtain high inductive propensity by inflammatory and IFN signaling [47,48,49,50,51,52,53,54]. Therefore, in addition to the ACE2 structure and affinity to S-RBD, the epigenetic evolution for IL-6 and ACE2 stimulation (reflected by higher mPWM scores), may serve as epigenetic biomarkers (or triggers) for susceptibility predictions of COVID-19 and other ARDS longitudinally across vertebrates and horizontally in subgroups of humans [30,47,48,49,50,51,52,53,54].

### 3.5. Overall Comparison of Phylogenic Topologies between IL-6 and ACE2 Gene Promoter Sequences

In addition to focusing on epigenetic analysis of these non-ISG CREs, we also conducted cross-species comparisons of phylogenic topology between the full proximal promoter sequences of IL-6 and ACE2 genes. Overall, the topology of the phylogenies of IL-6 and ACE2 gene promoters are similar, with a comparative topological score of 86.5% (Figure 7). Sharing a root of low vertebrates (*D. rerio* and/or *X. tropicalis*), the CoV2(+) species were distributed within the clades containing primates, carnivores, and glires. In contrast, all the ruminant promoters were clustered into a most phylogenically distant clade and associated with no CoV2(+) species (Figure 7). Comparison of the two phylogenies in detail showed that the major difference came from the location of the chicken, rabbit, guinea pig, and pig. In the IL-6 promoter phylogeny (Figure 7, left panel), chicken IL-6 promoter seems to derive rodent IL-6 gene promoters after evolution from the fish and frog; in the ACE2 promoter phylogeny (Figure 7, right panel), however, the chicken ACE2 promoter serves as a root leaf with the zebrafish. The largest difference is between the phylogenic positions of IL-6 and ACE2 gene promoters for pigs and guinea pigs. Whereas in the IL-6 promoter phylogeny, the porcine one sisters to those of the alpaca and horse, within the carnivore clade that contains most of the validated CoV2(+) species in addition to the primate clade, the porcine ACE2 gene promoter was next to the ruminant clade that has no CoV2(+) species identified so far [20,21,22,23,24,25,26,27,28,29]. Guinea pig as a rodent species has its IL-6 promoter surprisingly within the primate clade, but its ACE2 promoter appears more primitive and shares the clade with the frog. Given that the primate and carnivore clades contain most identified CoV2(+) species, if pig and guinea pig are proven to be CoV2(+) species, the IL-6 promoter phylogeny may better correlate to CoV2(+) prediction; otherwise, the ACE2 promoter phylogeny correlates better. The rabbit and otter, which occupy similar positions in both IL-6 and ACE2 promoter phylogenies, may have a high potential to be CoV2(+) and COVID-19-susceptible based on this and previous studies, which used epigenetic and structural models, respectively [30,31,32,33]. In this regard, pigs and guinea pigs may serve as symbol species to estimate the epigenetic role of non-ISGs in CoV2(+) prediction. No study has tested CoV2/COVID-19 susceptibility in guinea pigs, but studies in pigs concluded that the species was unsusceptible [20]. This may indicate that the overall epigenetic feature of ACE2 genes better relates to CoV2(+) status in some mammalian species. However, the study of key CRE scores of non-ISGs in Figure 6 indicates that IL-6 gene CRE scores have a higher correlation when compared between the CoV2(+) and CoV2(−) species. This may reflect an etiological fact that CoV2(+) is necessary but not sufficient for COVID-19 progression; and the latter is indeed dependent on the host immune reaction, particularly the early ISGs and non-ISG responses studied here [51,54]. In that regard, epigenetic evolution/regulation of ACE2 and IL-6 genes may signify two layers of COVID-19 progression, i.e., ACE2 is better for CoV2(+) and IL-6 is better for downstream COVID-19 symptoms [51,54,58,59].

### 3.6. Non-Bias Transcriptome-Based Categorization of Non-ISGs That Resemble to the Inductive Pattern to IL-6 or ACE2 Genes

Compared with canonical ISGs, studies of epigenetic regulation and expression of non-ISGs have just started accompanying our understanding of their role in some autoimmune and inflammatory diseases [50,51,52,53,54]. Although some non-canonical signaling pathways, that are independent of the canonical IFN-JAK-ISGF3 axis, play a role in ISG induction, the classification criteria of non-ISGs are not established [50,51,52,53,54]. Using IL-6 and ACE2 genes as examples of non-ISGs, the disparity of their cross-response to inflammatory and IFN signaling could be one way to classify them as IL-6-like or ACE2-like groups. We therefore analyzed a non-biased transcriptome (RNA-Seq) dataset from porcine alveolar macrophages treated with different stimuli and infected with a porcine arterivirus, a respiratory virus belonging to Nidovirales with coronaviruses [57]. We chose to use porcine transcriptome data because of the species-focus of our projects and the anatomical and physiological resemblance between pigs and humans [57]. Figure 8 presents the IL-6-like and ACE2-like groups, which were categorized based on their responsive patterns to liposaccharide (LPS) and two types of IFNs (i.e., IFN-α or type I and type II IFN-γ) at the early phase of 5 h post-treatment/infection [57]. These clustered IFN responsive genes were mainly from the RAAS, TNF, IL-6, chemokine superfamilies. For the IL-6 non-ISG group, all of these genes showed robust stimulation by LPS as well as a weaker response to both IFNs (Figure 8A). In contrast, the ACE2-group genes were insensitive to LPS, but were upregulated significantly by both types of IFNs (Figure 8B). Compared with the canonical group of ISGs (Figure 8C), which shows the highest response to the type I IFN-α, the IL-6 group had the least increase upon IFN-α and a similar stimulation by IFN-γ as for ISGs; and the ACE2 group showed a mid-response to IFN-α but highest to IFN-γ (Figure 8A–D). Figure 8D statistically demonstrates the stimulatory difference among three groups of IFN-responsive genes: (1) for ISGs: IFN-α > IFN-γ > LPS with a higher background expression in PBS, IL-4, and IL-10 treatments; (2) for IL-6-like non-ISGs: LPS > IFN-γ > IFN-α with the lowest background expression; and (3) for ACE2-like non-ISGs: IFN-γ > IFN-α > LPS with a mid-background expression. Therefore, our classification of ISGs and non-ISGs represents a complete scenario of gene response levels (i.e., at low, mid, and high levels of responses to LPS and two types of IFNs) to complement each other per their responsive propensity to LPS, IFN-γ, and IFN-α. As previously described, most ISGs, especially non-ISGs, are inter-regulated through multiple canonical and non-canonical signaling pathways. The cross-talking of signaling pathways mediated by different types of IFNs and inflammatory cytokines is dynamic to form into an intricate regulatory network underlying animal immunity to determine disease pathogenesis in various situations [50,51,52,53,54]. Therefore, with the functional extension of physiological genes such as AGT and ACE2, the new discovery of species-dependent response to viral infections and IFN stimulation posits them as immunogenetic factors critical to determining COVID-19 disease progression in addition to its role as a major virus receptor [44,45,46,47,48,49]. Notably, several ACE2 isoforms have been identified in humans and several major livestock species [30,60,61]. Our transcriptome analysis also picked up one short porcine ACE2 isoform (ACE2S)—its expression pattern is actually more like IL-6 non-ISGs than the consensus ACE2 longer isoform (ACE2L) [30]. In addition to ACE2, the AGT gene of the RAAS also showed a non-ISG property similar to ACE2 (Figure 8A,B). Collectively, transcriptomic annotation afforded us to cluster tentative non-ISGs that share expression patterns similar to IL-6 or ACE2 genes. Interestingly, most of them belong to IL-6, TNF, and chemokine superfamilies, whose roles in regulation of autoimmune and inflammatory diseases, as well as in COVID-19 progression, warrant further investigation.

## 4. Conclusions

Figure 9 depicts the working summary of this study for epigenetic evolution and regulation of IL-6 and ACE2 as non-ISGs, indicating their potentials as biomarkers for inflammatory syndrome underlying pathogenic viral infection such as of COVID-19. Non-ISGs such as those categorized by resemblance to IL-6 and ACE2 genes were sequentially regulated by TNF, IFN and TLR signaling, which modify chromatin accessibility through activating histone modification and recruitment of transcription factors including PU.1, IRF, and NF-κB binding on the promoter regions of these non-ISGs. In turn, it will amplify the inflammatory loop through IL-6-mediated response and inducing more ACE2 expression, which collectively contributes to the occurrence of respiratory and inflammatory syndromes as in COVID-19. Therefore, high expression of non-ISGs such as IL-6 and ACE2 could be biomarkers for the exacerbation of inflammation underlying some viral infections, especially those such as SARS-CoV-2, which dysregulates the physiological function of ACE2 in the RAAS-centric body systems. In addition, the cross-species epigenetic evolution of these key physio-pathological genes may provide a key to decipher molecular mechanisms underlying species-specific susceptibility to COVID-19 from the host side.

## Figures and Tables

**Figure 1 genes-12-00154-f001:**
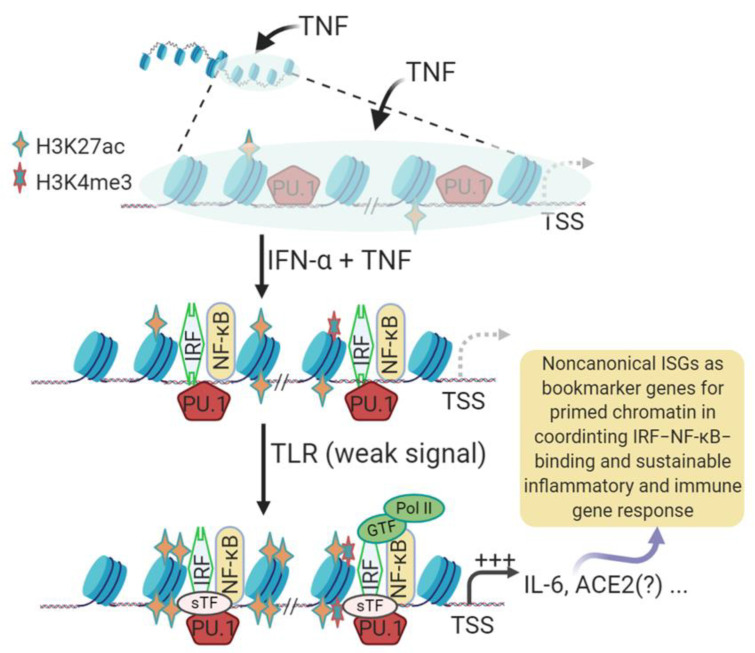
Schematic of epigenetic regulation and interferon (IFN) signaling to coordinate induction of non-canonical IFN-stimulated genes (non-ISGs). Stimulation of lung macrophages and epithelial cells with tumor necrosis factor (TNF) induces transient expression of TNF-target genes encoding inflammatory mediators, such as IL6 and TNF, followed by an insensitive state in which signaling responses to TLR ligands are strongly suppressed, and chromatin is not activated (depicted by a grey shade). This transient suppression state can be activated by a co-stimulation with TNF plus IFN-α and results in an increase in positive histone markers (mostly H3K4me3 and H3K27ac) and chromatin accessibility, which further coordinate binding of IRFs and NF-κB transcription factors and lead to non-ISG marker gene (such as IL-6) expression. Many inflammatory genes, including angiotensin converting enzyme 2 (ACE2), can be among these genes, which are bookmarked with primed chromatin and subsequently exhibit a robust transcriptional response even to very weak proximal TLR-induced signals, which may comprise a critical factor in the exacerbation of pulmonary inflammatory and COVID-19 syndrome. Adapted and redrawn from Barrat et al. (2019) [51]. Abbreviations: ac, acetyl; me, methylation; Pol, polymerase; PU.1, transcription factor binding to the PU-box, a.k.a SPI1; Non-ISG, non-canonical interferon stimulated genes; GTF, sTF, or TF, general (G), tissue-specific (s) transcription factor (TF); TLR, toll-like receptor; TSS, transcription start site.

**Figure 2 genes-12-00154-f002:**
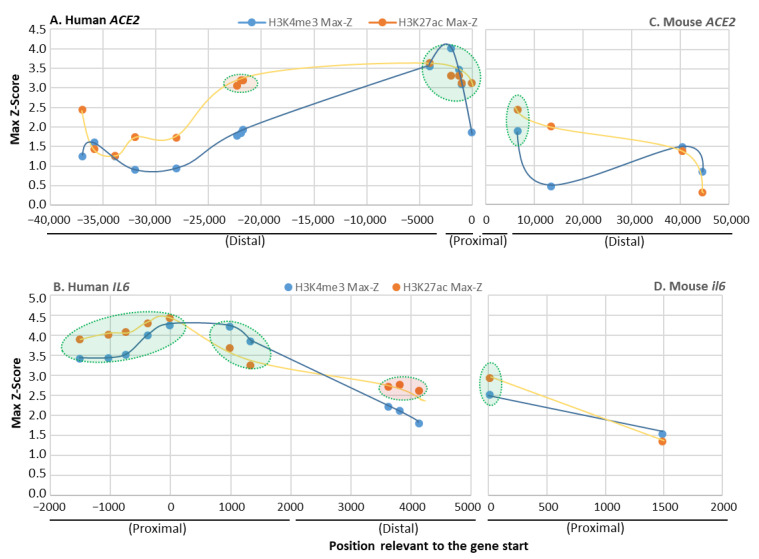
Profiling of positive histone markers (H3K4me3 and H3K27ac) indicating chromatin accessibility of RNA polymerase II adjacent to human and mouse ACE2 and IL-6 gene bodies, respectively. Annotation of ENCODE epigenetic datasets (Chip-Seq and ATAC-Seq from 839 and 157 cell/tissue types of humans and mice, respectively, from https://www.encodeproject.org/). Comparative existence of H3K4me3 and H3K27ac markers was detected between IL-6 and ACE2 gene promoters in either humans (**A**,**B**) and mice (**C**,**D**); however, higher Z-scores and enrichment of H3K4me3 and H3K27ac were found in human IL-6 and ACE2 genes (**A**,**B**) than their orthologs in mice (**C**,**D**). Distal, >2000 bp before the transcription start sites (TSS), and proximal promoter within 2000 bp before the TSS. Datasets with Z-score higher than the overall average are shaded with oval shapes.

**Figure 3 genes-12-00154-f003:**
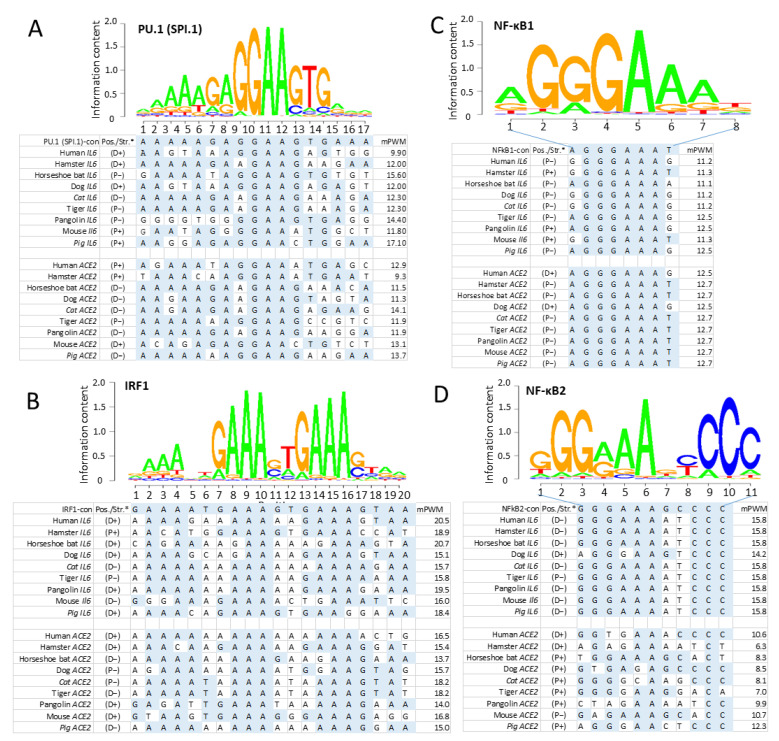
Existence of *cis*-regulatory elements (CREs) that bind typical non-ISGs transcription factors of (**A**) PU.1 (a.k.a. SPI1), (**B**) IRF1, and (**C**,**D**) NF-κB1/2 in the promoter regions of IL-6 and ACE2 gene orthologs from the representative two SARS-CoV-2-unsusceptible species (pigs and mice) and seven susceptible species. All three types of CREs have comparable Log_2_(mPWM) scores between ACE2 and IL-6 genes, except NF-κB2 that mediates non-canonical NF-κB response. (**D**) has a significant lower mPWN score (2–6 Log_2_ units), indicating ACE2 genes are among different non-ISGs group other than IL-6. P/D, proximal or distal regions of promoters; ± sense or antisense strands. mPWM scores were calculated using tools at https://ccg.epfl.ch/pwmtools/pwmscore.php with CRE Matrices are from MEME-derived HOCOMOCOv11 TF collection affiliated with the PWM tools. PWM, position weight matrix.

**Figure 4 genes-12-00154-f004:**
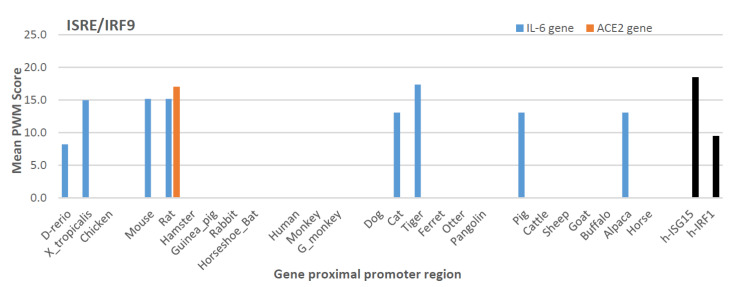
Lack of ISRE/IRF9 binding site that responds to IFN signaling for ISG expression in analyzed IL-6 and ACE2 genes. Cross-species analysis of mean PWM (mPWM) scores of *cis*-regulatory elements (CREs) that bind ISRE/IRF9 in the proximal promoter regions of IL-6 and ACE2 gene orthologs from the 25 representative vertebrate species. mPWM score is presented in a Log_2_(mPWM) scale. It further indicates that IL-6 and especially ACE2 genes in most species are non-ISGs. Canonical ISGs of human ISG15 and IRF1 are used as references (black bars). mPWM scores were calculated using tools at https://ccg.epfl.ch/pwmtools/pwmscore.php with CRE matrices from MEME-derived HOCOMOCOv11 TF collection affiliated with the PWM tools. PWM, position weight matrix. Abbreviations: D-rerio, Danio rerio (Zebrafish); X_trapicalis, Xenopus trapicalis; G_monkey, African Green Monkey; h-, human.

**Figure 5 genes-12-00154-f005:**
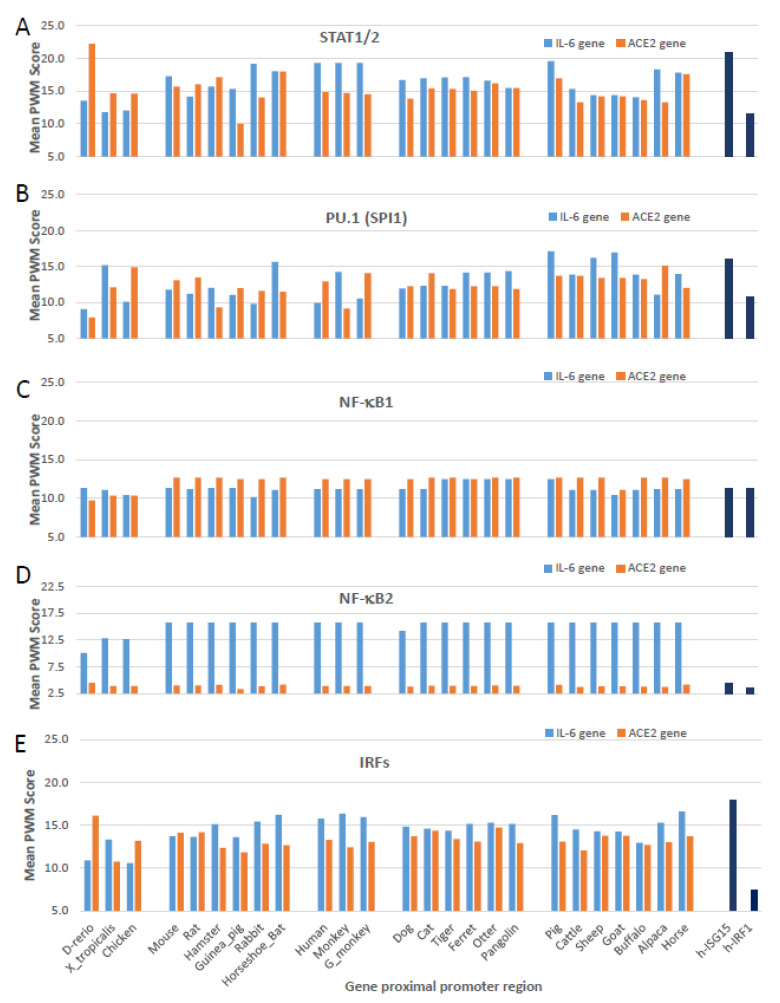
Cross-species analysis of mean PWM scores of *cis*-regulatory elements (CREs) that bind (**A**) STAT1/2, (**B**) PU.1 (a.k.a. SPI1), (**C**) NF-κB1, (**D**) NF-κB2, and (**E**) IRFs (including IRF1-9, which show significant PWM scores with *p* < 0.0001) in the proximal promoter regions of IL-6 and ACE2 gene orthologs from the 25 representative vertebrate species. All types of CREs have comparable Log_2_(mPWM) scores between ACE2 and IL-6 genes, except NF-κB2 that mediates non-canonical NF-κB response. (**D**) has a significant lower mPWN score (2–6 Log_2_ units), indicating ACE2 genes are among different non-ISGs group other than IL-6. Canonical ISGs of human ISG15 and IRF1 are used as references. mPWM scores were calculated using tools at https://ccg.epfl.ch/pwmtools/pwmscore.php with CRE matrices from MEME-derived HOCOMOCOv11 TF collection affiliated with the PWM tools. PWM, position weight matrix. Other abbreviations are as in Figure 4.

**Figure 6 genes-12-00154-f006:**
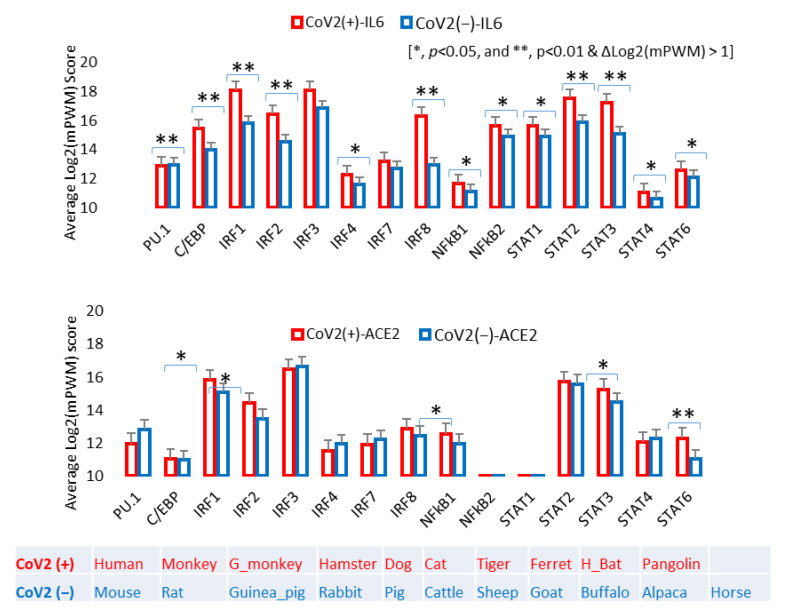
Cross-species correlation of epigenetically regulatory CREs, which associate with inflammatory and IFN signaling, in IL-6 and ACE2 gene promoters as biomarkers for COVID-19 susceptibility. Mean PWM (mPWM) scores were generated as described in previous figures, and compared between two groups of known COVID-19 susceptible species [CoV2(+)] and unsusceptible species [CoV2(−)]. This shows that ACE2, and especially IL-6 genes, from CoV2(+) species contain the CREs which have significantly higher mPWM scores, indicating that in some vertebrate species, non-ISGs such as ACE2 and especially IL-6 genes evolved to obtain high inductive propensity by inflammatory and IFN signaling, and may serve as epigenetic biomarkers (or triggers) for susceptibility prediction of COVID-19 and other ARD syndrome. *, *p* < 0.05, and **, *p* < 0.01 and ∆Log_2_(mPWM) > 1; n = 10, compared between the CoV(+) and CoV(−) groups. Abbreviation: H_Bat, great horseshoe bat; other abbreviations are as in Figure 4.

**Figure 7 genes-12-00154-f007:**
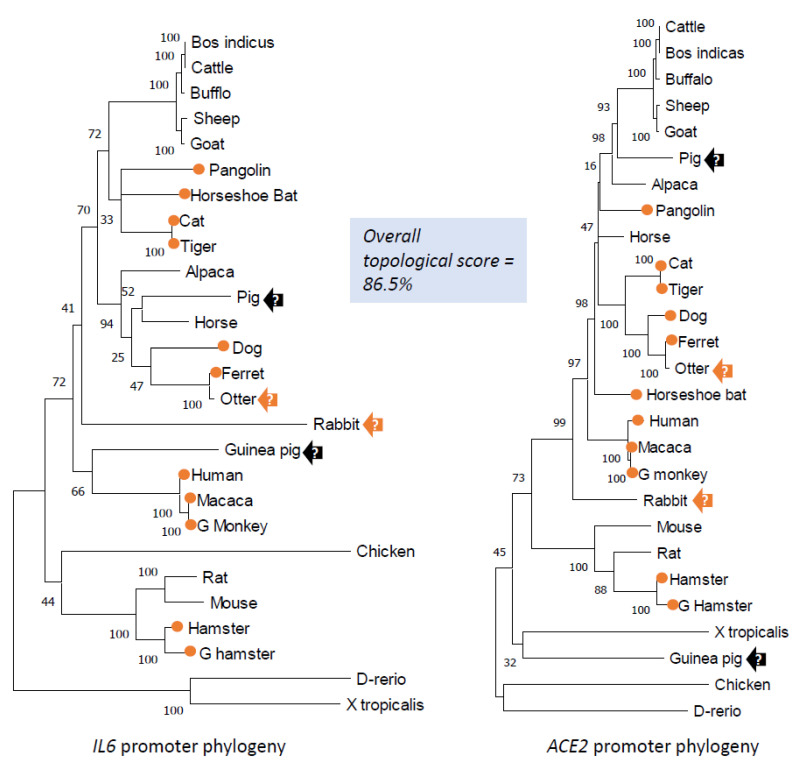
Cross-species phylogenic and topological comparison of IL-6 and ACE2 gene promoters. Evolutionary analyses were conducted in MEGA X. The evolutionary history was inferred by using the maximum likelihood method and Tamura–Nei model. The tree with the highest log likelihood (−52,755.39) is shown. The percentage of trees in which the associated taxa clustered together is shown next to the branches. Initial tree(s) for the heuristic search were obtained automatically by applying Neighbor-Join and BioNJ algorithms to a matrix of pairwise distances estimated using the Tamura–Nei model, and then selecting the topology with superior log likelihood value. For topological comparison between phylogenic trees generated using IL-6 and ACE2 gene proximal promoters, the phylogenies of Newick strings were generated using MEGA, and topological comparison between the Newick trees was performed with Compare2Trees at (http://www.mas.ncl.ac.uk/~ntmwn/compare2trees) to obtain the overall topological scores. Orange circle: COVID-19 susceptible species. Arrows: other tentative marker species to determine which group (IL-6 or ACE2) of non-ISGs are more determined for COVID-19 susceptibility. Abbreviations are as in Figure 4.

**Figure 8 genes-12-00154-f008:**
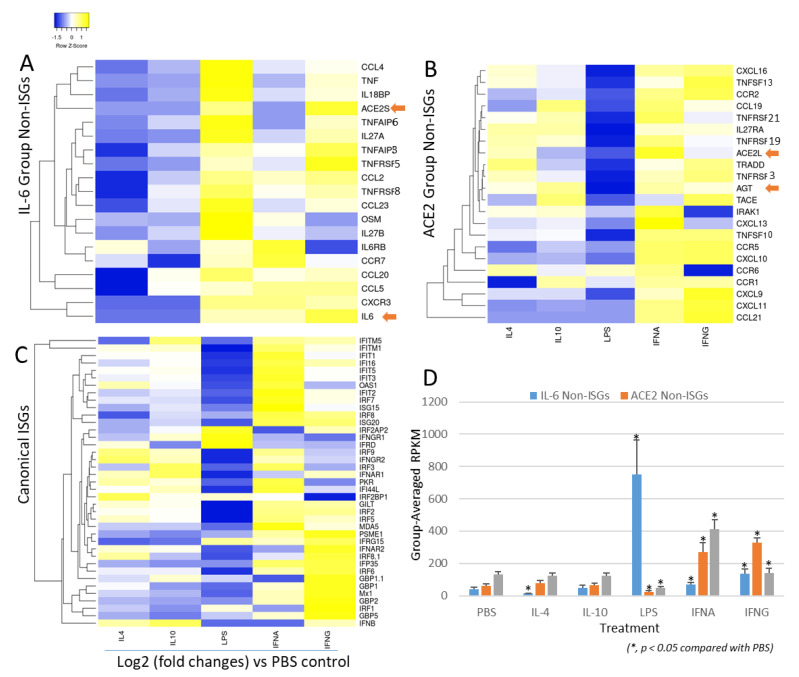
Genome-wide categorizing non-ISGs based on the similarity of inductive pattern to IL-6 and ACE2 genes. The non-biased genome-wide transcriptomic data were generated using an RNA-Seq procedure in porcine lung macrophages stimulated with each activation stimulator of IL-4, IL-10, LPS, IFN-α or IFN-γ at 20 ng/mL and infected by porcine arterivirus virus for 5 h, using an Illumina procedure as previously described [57]. Significantly differentially expressed genes (DEGs) in the renin–angiotensin system (RAS), interleukin (IL)-6, TNF, and chemokine super-families were annotated and grouped using heatmaps according to their inductive expression patterns similar to: (**A**) IL-6, (**B**) ACE2; (**C**) Examples of canonical ISGs as reference; (**D**) Averaged transcriptomic expression levels (normalized at reads per kilobase of transcript per million mapped reads, RPKM) of the grouped ISGs or non-ISGs above. Indicated by arrows, pigs have two ACE2 isoforms, namely ACE2L and ACE2S, which have different expression patterns; ACE2S similar to IL-6 showed to be less responsive to IFN-α but highly responsive to LPS and IFN-γ. In contrast, ACE2L and another key gene, AGT, in RAS were categorized together with other non-ISGs (**B**), which is more like the expression pattern of canonical ISGs (**C**) than the IL-6 group (**A**).

**Figure 9 genes-12-00154-f009:**
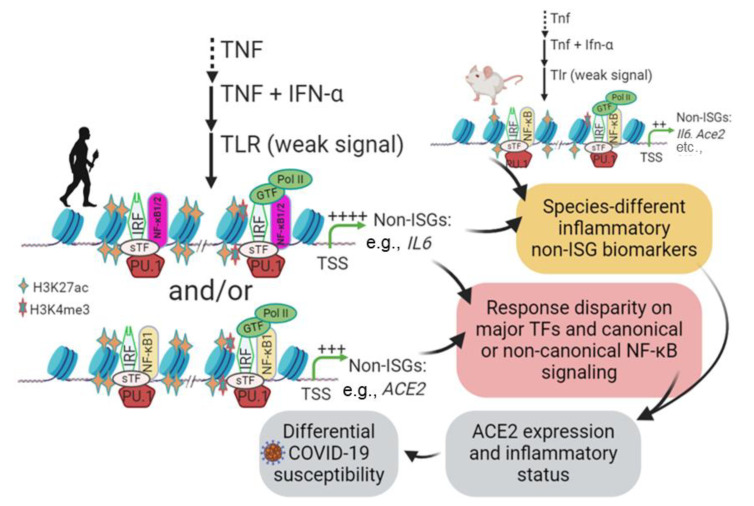
Working summary for IL-6 and ACE2 as non-ISGs biomarkers and contribution to COVID-19 susceptibility. Epigenetic regulation of non-ISGs such as IL-6 and ACE2 was sequentially regulated by TNF, IFN and TLR signaling, which modify chromatin accessibility through activating histone modification and recruitment of transcription factors including PU.1, IRF and NF-κB binding on promoter regions of IL-6 and ACE2 genes. In turn, it will amplify inflammatory loops through IL-6-mediated responses and induce more ACE2 expression, which collectively contributes to the occurrence of respiratory distress syndrome, as in COVID-19. Therefore, high expression of non-ISGs such as IL-6 and ACE2 could be biomarkers to determine COVID-19 susceptibility and disease development in different animal species. Abbreviations: non-ISG, non-canonical interferon stimulated genes; GTF, sTF, or TF, general (G), tissue-specific (s) transcription factor (TF); TLR, toll-like receptor; TSS, transcription start site.

## Data Availability

The porcine RNA-Seq dataset is available at https://trace.ddbj.nig.ac.jp/DRASearch/study?acc=SRP033717 for NCBI BioProject with an accession number of SRP033717.

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
