# Peer review of "Epigenetic Evolution of ACE2 and IL-6 Genes: Non-Canonical Interferon-Stimulated Genes Correlate to COVID-19 Susceptibility in Vertebrates"

_genes, 2021, doi:10.3390/genes12020154_

Round 1

Reviewer 1 Report

Initially, I congratulate colleagues for the excellent manuscript. Because of the seriousness of the pandemic caused by the new Coronavirus, the present manuscript contains information of vital importance for public health worldwide. The authors provide valuable information on the epigenetic correlation and susceptibility of COVID-19. The study evaluated the genes of the enzyme ACE2, and interleukin (IL) and showed evidence in the active epigenetic evolution through 19 histones modification and interaction of cis/trans factors in different species of vertebrates. Also, the information obtained by the authors on the epigenetic properties of the ACE2 and IL-6 28 genes can serve as biomarkers to longitudinally predict the susceptibility of COVID-19 in vertebrates and partially explain the inequality of COVID-19 in people from different subgroups. This rapporteur, when reading this manuscript, can feel the modifying power of science. The summary brings enthusiasm and longing for the complete reading of the manuscript. The introduction is well-founded and brings clarity to the reader, the methodology is robust and the results bring important scientific postulates. And finally, the discussion and conclusion that underlies the findings and brings mental anxiety to the reader who eagerly awaits the new chapters of this heroic and scientific journey. For true heroes do not wear capes, but lab coats. In this way, this rapporteur would like as soon as possible to share these findings with the world and approve this manuscript for publication as it stands. And he wishes enlightenment to the noble colleagues for the next events.

Author Response

We highly appreciate for these positive and encouraging comments.

Reviewer 2 Report

Sang et al. conducted an extensive study for epigenetic evolution and regulation of IL-6 and ACE2 as non-ISGs in correlation to COVID-19 susceptability using both open source data and experimental data. The paper is well written and nicely structured. This reviewer thinks that the results and discussion section is unproportionally long in compared with the method section. Authors may think about shortening it for the ease of reading. In contrast, the RNA-seq and data analysis section is very brief. Details about the NCBI RNA-seq data are missing and more details can be provided concerning data analysis. Was it a simple two-group comparison using t-test or linear regression? As age and sex are important factors for susceptability and clinical outcomes, were they inlcuded in statistical analysis for adjustment? 

Overall, I find the paper interesting and informative to readers of Genes.

Author Response

Sang et al. conducted an extensive study for epigenetic evolution and regulation of IL-6 and ACE2 as non-ISGs in correlation to COVID-19 susceptibility using both open source data and experimental data. The paper is well written and nicely structured. This reviewer thinks that the results and discussion section is unproportionally long in compared with the method section. Authors may think about shortening it for the ease of reading.

We thank you for this positive and constructive comment. Proofreading were carefully performed during the revision to improve the readability and a proportional balance as suggested. We would like to point out that the combinative format of the Results and Discussion Sections, which include nine compound figures, thus is reasonable to take about 50% (11 of 21 pages) of total page capacity.  

In contrast, the RNA-seq and data analysis section is very brief. Details about the NCBI RNA-seq data are missing and more details can be provided concerning data analysis. Was it a simple two-group comparison using t-test or linear regression? As age and sex are important factors for susceptibility and clinical outcomes, were they included in statistical analysis for adjustment?

During cross-species annotation (especially for promoter regions) of ACE2 and IL-6 genes, RNA-Seq datasets that are affiliated to NCBI gene entries (such as BioProjects PRJEB4337 and PRJNA66167 for humans and mice genes) were used to verify the gene expression per RNA-Seq exon/intron coverage analyses. The detailed records for NCBI RNA-Seq data analyses was provided as in Supplement Excel Sheet. We particularly re-analyzed our own RNA-Seq data sheets (BioProject  SRP033717) for pilot categorizing IL-6 and ACE-2 like non-ISGs as presented in Figure 8. The detailed information about the generation and quality control analysis of the RNA-Seq dataset was described in Ref [57] and referred here. In addition, it is a dataset from a six-group comparison using typical RNA-Seq Illumina bioinformatic pipeline for significant and regression tests [57]. The sample resource are primary alveolar macrophages without incorporation of the age and sex factors at this stage. However, we appreciate for this insightful and foreseeing suggestion for the future endeavor along with this direction.

 The following description was added on Line 212-216:

 “During cross-species annotation of ACE2 and IL-6 genes, RNA-Seq datasets that are affiliated to NCBI gene entries (such as BioProjects PRJEB4337 and PRJNA66167 for humans and mice genes) were used to verify the gene expression per RNA-Seq exon/intron coverage analyses. The detailed records for NCBI RNA-Seq data analyses was provided as in Supplement Excel Sheet.”

Overall, I find the paper interesting and informative to readers of Genes.

We appreciate for this concise and positive comment.

Reviewer 3 Report

Ace2 and IL-6 play major role in SARS-Cov19 pathogenesis. Sang et al identified these genes as non-canonical interferon-stimulated genes. They employed comparative genomics and analysis of regulation of their expression to conclude that there is evolutionary evidence of correlation between epigenetic status of these loci and species susceptibility to SARS-Cov19.

Overall, paper is well written, conclusions are logical and fully supported by the analysis of publicly available data as well as by investigator’s own data.

However, I would like to point out some serious differences between IL-6 and ACE2 genes that affect their regulation and were overlooked by the authors.

  1. IL-6 is a short gene located in gene desert and regulated by distal superenhancers. It has CTCF site exactly at the promoter. This site is adjacent to a myriad of other transcription binding sites. Therefore IL-6 promoter resembles enhancers in its histone marks – very high level of H3K27Ac.
  2. ACE2 is a relatively long gene surrounded by other genes. It’s major promoter has only minor H3K27Ac and binds very few transcription factors than IL-6 promoter. There are several ACE2 transcripts and one of them is originating around Exon 9. There is a much stronger isle of H3K27Ac around this region (GRCh37/hg19 Assembly) than at the regular promotor. Also there is a LTR element within ACE2 gene body which serves as a promoter.

Therefore, I would like the authors to discuss these peculiarities as well as these two references in the Discussion.

A novel ACE2 isoform is expressed in human respiratory epithelia and is upregulated in response to interferons and RNA respiratory virus infection.

Blume C, Jackson CL, Spalluto CM, Legebeke J, Nazlamova L, Conforti F, Perotin JM, Frank M, Butler J, Crispin M, Coles J, Thompson J, Ridley RA, Dean LSN, Loxham M, Reikine S, Azim A, Tariq K, Johnston DA, Skipp PJ, Djukanovic R, Baralle D, McCormick CJ, Davies DE, Lucas JS, Wheway G, Mennella V.Nat Genet. 2021 Jan 11. doi: 10.1038/s41588-020-00759-x. Online ahead of print.PMID: 33432184

Tissue-specific and interferon-inducible expression of nonfunctional ACE2 through endogenous retroelement co-option.

Ng KW, Attig J, Bolland W, Young GR, Major J, Wrobel AG, Gamblin S, Wack A, Kassiotis G.Nat Genet. 2020 Dec;52(12):1294-1302. doi: 10.1038/s41588-020-00732-8. Epub 2020 Oct 19.PMID: 33077915

There are several minor corrections that should be made:

Line 40 – Mortal rate to mortality rate

Lines 49-52 I think authors push term “epigenetic regulation” too wide to include what is just good old regulation of gene transcription.

243 – non-ISGs is not activated – should be plural

245 – positive histone markers – change to more common - marks

261 - defined as active enhancer markers - H3K4me3 – active promoters and enhancers, H3K27ac – enhancers and promoter proximal regions. I think this description is not very good. Need to be more specific - H3K4me3 and H3K27ac are highly enriched at active promoters near the transcription start site (TSS) and positively correlated with transcription. H3K27ac is also an active enhancer mark.

271 - fold more of these positive histone medication – plural and modification

275 - modification, H3K4me3 and H3K27ac, is significantly associated – plural

299 – primal – change to primary

308 - 2-8 Log2units – add space

317 – differ – better word would be distinguish

340 – black bars (describe what they are in the figure legend)

404 - the CREs that have significantly higher (remove that)

Author Response

Ace2 and IL-6 play major role in SARS-Cov19 pathogenesis. Sang et al identified these genes as non-canonical interferon-stimulated genes. They employed comparative genomics and analysis of regulation of their expression to conclude that there is evolutionary evidence of correlation between epigenetic status of these loci and species susceptibility to SARS-Cov19.

Overall, paper is well written, conclusions are logical and fully supported by the analysis of publicly available data as well as by investigator’s own data.

We appreciate for these positive and encouraging comments.

 However, I would like to point out some serious differences between IL-6 and ACE2 genes that affect their regulation and were overlooked by the authors.

IL-6 is a short gene located in gene desert and regulated by distal super enhancers. It has CTCF site exactly at the promoter. This site is adjacent to a myriad of other transcription binding sites. Therefore IL-6 promoter resembles enhancers in its histone marks – very high level of H3K27Ac.

ACE2 is a relatively long gene surrounded by other genes. It’s major promoter has only minor H3K27Ac and binds very few transcription factors than IL-6 promoter. There are several ACE2 transcripts and one of them is originating around Exon 9. There is a much stronger isle of H3K27Ac around this region (GRCh37/hg19 Assembly) than at the regular promotor. Also there is a LTR element within ACE2 gene body which serves as a promoter.

Therefore, I would like the authors to discuss these peculiarities as well as these two references in the Discussion.

A novel ACE2 isoform is expressed in human respiratory epithelia and is upregulated in response to interferons and RNA respiratory virus infection.

Blume C, Jackson CL, Spalluto CM, Legebeke J, Nazlamova L, Conforti F, Perotin JM, Frank M, Butler J, Crispin M, Coles J, Thompson J, Ridley RA, Dean LSN, Loxham M, Reikine S, Azim A, Tariq K, Johnston DA, Skipp PJ, Djukanovic R, Baralle D, McCormick CJ, Davies DE, Lucas JS, Wheway G, Mennella V.Nat Genet. 2021 Jan 11. doi: 10.1038/s41588-020-00759-x. Online ahead of print.PMID: 33432184

Tissue-specific and interferon-inducible expression of nonfunctional ACE2 through endogenous retroelement co-option.

Ng KW, Attig J, Bolland W, Young GR, Major J, Wrobel AG, Gamblin S, Wack A, Kassiotis G.Nat Genet. 2020 Dec;52(12):1294-1302. doi: 10.1038/s41588-020-00732-8. Epub 2020 Oct 19.PMID: 33077915

 We especially appreciate this professional comment. Yes, a much higher Z-Score of H3K27Ac was detected in human IL-6 gene than ACE2 as presented in Figure 2 to support this fact. Still it is an unsolved issue about whether this observation is conserved on all vertebrates IL-6/ACE2 genes as not applicable to mice. We have noted the isoform diversity of ACE2, and their expression and functional difference upon interferon stimulation and SARS-CoV2 susceptibility. In our previous publication (Ref. 30), we analyzed the evolution of this kind of short ACE2 isoforms in vertebrates and proposed their potential role in the tolerance of some vertebrate species to SARS-CoV2 infection. Here we focus on the more prevalent (i.e. more cross-species conservative) long ACE2 isoform for promoter/epigenetic analysis and accordingly categorize the ACE2 and IL-6 as different groups of non-canonical interferon stimulated genes (non-ISGs). The long ACE2 isoform representing a cross-species consensus isoform, which alternatively serves as a viral receptor for SARS-CoV2 infection in addition to its pivotal role in RAAS, is not a canonical ISG stimulated primarily by antiviral interferons, but a non-ISG needs a sequential (or combinative) epigenetic regulation as seen in progressive COVID-19 patients.  The short ACE2 isoform, as revealed in our previous publication and the two recent ones as suggested above, is hypothesized to conserve mostly ACE2’s enzymatic function in RAAS but is truncated the SARS-spike binding domain to void the virus binding. This together with its IFN-responsive LTR-containing promoter potentially provides an evolutionary advantage to retain RAAS physiological function when the virus binds and suppresses the physiological function of the long ACE2 isoform. However, a perplexing issue directly aroused by the discovery about the short human ACE2 isoform is that, its IFN-stimulation has little application (“non-functional”) in suppression of SARS-CoV2 infection in respiratory epithelial cells, indicating that the long ACE2 isoform is still expressed there to facilitate SARS-CoV2 prevalence as we observed. It is noteworthy that: (1) both studies cited above still only tested the classical IFN-stimulating property of either long or short ACE2 isoform using a single IFN treatment scenario; and (2) our findings here about the epigenetic stimulation (i.e. sequentially through inflammatory and IFN signaling) instead of simple IFN stimulation of long ACE2 isoform actually provides a rationale to explain the pathogenic development of COVID-19 progression. In addition, it needs to indicate that the IFN stimulating property of ACE2 genes are evolving and behaves differently among vertebrate species. For example, in Figure 8, we show that at least in porcine lung cells, the short ACE2 isoform (ACE2S) highly responds to both LPS- and IFNα-stimulations, and the long ACE2 isoform (ACE2L), by contrast, is more responsive to both types of IFN treatments.                        

The following discussion was add on Line 276-281:

“Notably, human IL-6 is a short gene located distantly from other coding genes and might correlated to higher histone marks (esp. H3K27Ac) relevant to distal super enhancers.  By contrast, human ACE2 gene is a relatively long gene surrounded by other genes, and its major promoter may be compacted in a more proximal region and has a limited H3K27Ac marks spanning the 2000-5000 bp distal region.”

And the two new references published during our submission was cited as:

  1. Ng KW, Attig J, Bolland W, Young GR, Major J, Wrobel AG, Gamblin S, Wack A, Kassiotis G. Tissue-specific and interferon-inducible expression of nonfunctional ACE2 through endogenous retroelement co-option. Nat Genet. 2020 Dec;52(12):1294-1302.
  2. Blume C, Jackson CL, Spalluto CM, Legebeke J, Nazlamova L, Conforti F, Perotin JM, Frank M, Butler J, Crispin M, Coles J, Thompson J, Ridley RA, Dean LSN, Loxham M, Reikine S, Azim A, Tariq K, Johnston DA, Skipp PJ, Djukanovic R, Baralle D, McCormick CJ, Davies DE, Lucas JS, Wheway G, Mennella V. A novel ACE2 isoform is expressed in human respiratory epithelia and is upregulated in response to interferons and RNA respiratory virus infection. Nat Genet. 2021 Jan 11. doi: 10.1038/s41588-020-00759-x. Epub ahead of print.

There are several minor corrections that should be made:

Line 40 – Mortal rate to mortality rate

Corrected.

Lines 49-52 I think authors push term “epigenetic regulation” too wide to include what is just good old regulation of gene transcription.

According to a recent definition, “Epigenetic processes regulate gene expression by modulating the frequency, rate, or extent of gene expression in a mitotically or meiotically heritable way that does not entail a change in the DNA sequence”. Epigenetic regulation thus may include the “phenotype” diversity resulted from differential transcription beyond changes of genetic code/composition. We add “except those have inborn genetic mutations” to clarify this point.

243 – non-ISGs is not activated – should be plural

Corrected. It is now on Line 247.

245 – positive histone markers – change to more common - marks

Changed. It is now on Line 249

261 - defined as active enhancer markers - H3K4me3 – active promoters and enhancers, H3K27ac – enhancers and promoter proximal regions. I think this description is not very good. Need to be more specific - H3K4me3 and H3K27ac are highly enriched at active promoters near the transcription start site (TSS) and positively correlated with transcription. H3K27ac is also an active enhancer mark.

Thank you, and it is changed to “as positive epigenetic marks enhancing relevant gene expression” in Line 265-266.

271 - fold more of these positive histone medication – plural and modification

Corrected. It is now on Line 274-275.

275 - modification, H3K4me3 and H3K27ac, is significantly associated – plural

Corrected. It is now on Line 283.

299 – primal – change to primary

Corrected. It is now on Line 308.

308 - 2-8 Log2units – add space

Added. It is now on Line 317.

317 – differ – better word would be distinguish

Changed. It is now on Line 326.

340 – black bars (describe what they are in the figure legend)

Described on Line 349.

404 - the CREs that have significantly higher (remove that)

Removed on Line 413.